# Assessment of Orthodontic Treatment Need and Oral Health-Related Quality of Life in Asthmatic Children Aged 11 to 14 Years Old: A Cross-Sectional Study

**DOI:** 10.3390/children10020176

**Published:** 2023-01-18

**Authors:** Adrián Curto, Fátima Mihit, Daniel Curto, Alberto Albaladejo

**Affiliations:** 1Pediatric Dentistry, Department of Surgery, Faculty of Medicine, University of Salamanca, Alfonso X El Sabio Avenue s/n, 37007 Salamanca, Spain; 2Faculty of Medicine, University of Salamanca, Alfonso X El Sabio Avenue s/n, 37007 Salamanca, Spain; 3Department of Pathology, 12 de Octubre University Hospital, Córdoba Avenue s/n, 28041 Madrid, Spain; 4Orthodontics, Department of Surgery, Faculty of Medicine, University of Salamanca, Alfonso X El Sabio Avenue s/n, 37007 Salamanca, Spain

**Keywords:** orthodontics, malocclusion, orthodontic treatment need, oral health-related quality of life, self-concept, adolescents

## Abstract

This study investigated the need for orthodontic treatment in asthmatic children aged 11 to 14 years and how the treatment affected their oral health-related quality of life (OHRQoL). Materials and Methods: This cross-sectional study was conducted at the dental clinic of the University of Salamanca in 2020–2022. The study selected a consecutive sample of 140 children with asthma (52.1% girls; 47.9% boys). This study used the Orthodontic Treatment Needs Index (OTN) to analyze the need for orthodontic treatment and the Children’s Perception Questionnaire (CPQ11–14) to assess OHRQoL. Results: Sex and age did not significantly influence the need for orthodontic treatment, although age may be considered influential for OHRQoL concerning oral symptoms (*p* < 0.01), functional limitations (*p* < 0.05), and total score on the CPQ_11–14_ questionnaire (*p* < 0.05): the younger the age, the greater the effect of the need for orthodontic treatment on OHRQoL. The social well-being of the patients was much more significantly impacted by the need for orthodontic treatment (15.7 ± 1.91) than by oral symptoms (7.64 ± 1.39), which were the least impacted. In all parts of the CPQ_11–14_ questionnaire and in the patients’ total scores, we observed significant agreement (*p* < 0.01) that treatment influenced OHRQoL. Conclusion: An inverse relationship exists between the severity of the treatment needed and OHRQoL.

## 1. Introduction

Asthma is a chronic respiratory childhood pathology that causes a considerable reduction in quality of life [1]. In Spain, approximately 10% of children are asthmatic, of whom 85% have an allergic etiology. The dominant symptoms of asthma-related airway obstruction are coughing, wheezing, and respiratory distress, but these vary with time and intensity. The origin of asthma is multifactorial [2].

In children and adolescents, asthma causes limitations in daily activity, physical activities, recreational activities, and school performance, and even interferes with sleep [3].

Asthmatic patients may also be at a higher risk of developing dental diseases. The main oral manifestations of asthmatic pediatric patients are the appearance of dental erosion, oral candidiasis, and increases in the prevalence of caries and gingival pathology. In part, the etiology of these manifestations is from the drugs taken to treat asthma. As the prevalence of asthma rises, asthma medication could cause considerable dental health problems worldwide [4,5].

The use of tools to assess malocclusions and their psychosocial impact has become increasingly common due to scholars recognizing the importance of measuring oral health-related quality of life (OHRQoL) when conducting clinical studies. This has led to an increase in the number and use of specific instruments to assess OHRQoL, which provides essential information to assess treatment needs [6,7].

The factors that influence a patient to start orthodontic treatment to correct a malocclusion are mainly dental and facial aesthetics. An awareness that facial appearance affects OHRQoL has increased the demand for orthodontic treatment [8]. According to the results of published studies, correcting malocclusions considerably increases OHRQoL in children [9,10].

Few studies have explored the influence of asthma on OHRQoL and the need for orthodontic treatment in children. According to the scientific literature, no statistically significant relationship exists between the two variables [11,12].

The main objectives of this study are to analyze the need for orthodontic treatment in asthmatic children aged 11 to 14 years (inclusive) and to evaluate their OHRQoL. The secondary objective is to study the influence of the sex and age of the participants on the main objectives. We hypothesized that asthmatic children with more severe malocclusions would have a worse OHRQoL.

The results of this study may contribute to the development of orthodontic preventive plans in the adolescent asthmatic population. Being able to treat dental malocclusions at an early stage is important as doing so will reduce their severity in the future and will allow scholars to know what possible dental malocclusions can be observed in the adolescent asthmatic population. In this case, the specialists will be able to know the OHRQoL of adolescent asthmatic patients.

To the best of our knowledge, no published studies evaluating the influence of orthodontic treatment on OHRQoL in asthmatic children exist.

## 2. Materials and Methods

### 2.1. Study Design

The present study was approved by the Bioethics Committee of the University of Salamanca (USAL 142/20). We report the data according to STROBE guidelines. The study was performed according to the 1964 Declaration of Helsinki and its later amendments [13].

The present study identified a sample of 140 consecutive children through the dental screening program at the dental clinic of the University of Salamanca between January 2020 and November 2022. The program consisted of an oral evaluation to determine whether orthodontic treatment was necessary. We informed the participants about the procedure and the confidentiality of the information obtained. We excluded patients who declined to participate. We obtained written informed consent from all subjects. A single examiner performed the oral examinations, determined the presence of malocclusions, assessed the severity of the malocclusions, and provided the patients with the oral quality of life questionnaire. Prof. Dr. Adrián Curto was the examiner, and he is a practitioner with a clinical focus on orthodontics and pediatric dentistry.

The inclusion criteria were as follows: patients aged between 11 and 14 years and diagnosed with asthmatic pathology according to the criteria established by the Spanish Society of Pediatrics (GEMA guide) [2,14]. To be included in the study, patients had to have been diagnosed with asthma at least one year before the study began. The exclusion categories were previous orthodontic treatment, caries or untreated periodontal pathology, craniofacial anomalies such as cleft lip or palate, symptoms or diagnosis of temporomandibular joint pathology, and cognitive disorders. We performed a detailed history to confirm that the participants met the inclusion and exclusion criteria.

### 2.2. Sample Size Calculation

Considering that the Salamanca region has an approximate population of 14,000 children between 11 and 14 years of age, we estimated a minimum sample size of 137 participants based on the incidence of asthma (10.0%), a confidence level of 5.0%, and a maximum error of ±5.0%. The number of participants could have risen to 151 if we added a 10.0% loss prevention. The sample size collected was 140, which met the estimated minimum [14].

### 2.3. Orthodontic Treatment Need Assessment

This study used the Index of Orthodontic Treatment Need (IOTN), initially described by Brook and Shaw in 1989 [15], to determine the orthodontic treatment needs of the patients. Scholars have verified the validity and reliability of the index in different countries and with different studies [16,17,18,19,20].

This index consists of two components: a clinical dental health component (DHC) and an aesthetic component (AC). We separately recorded the DHC and AC. The DHC registers the malocclusion characteristics, and the AC has 10 grades, corresponding to 10 color photographs showing different levels of dental orthodontic attractiveness from 1 (best appearance) to 10 (worst appearance). The clinicians matched one of the images to the patient’s malocclusion. Depending on the pathology, we included each participant in one of the five levels: Grade 1 (normal occlusion or mild malocclusion—no need for orthodontic treatment), Grade 2 (mild malocclusion—small need for orthodontic treatment), Grade 3 (moderate malocclusion—limited need for orthodontic treatment), Grade 4 (severe malocclusion––need for treatment), and Grade 5 (very severe malocclusion—need for treatment) [15].

### 2.4. Oral Health-Related Quality of Life Assessment

The present study measured OHRQoL using the Spanish version of the Child Perceptions Questionnaire developed by Jokovic et al. in 2002 (CPQ-Esp_11–14_) [21,22]. This questionnaire was the first instrument to be used to analyze OHRQoL in children. The CPQ_11–14_ questionnaire covers the following categories: oral symptoms, functional limitation, emotional well-being, and social well-being. This questionnaire consists of 37 questions and has a recall time of three months. We recorded responses on a Likert scale: 0 = never, 1 = once or twice, 2 = sometimes, 3 = often, and 4 = every day or almost every day. The responses had a possible range from 0 to 103: the higher the score, the worse the quality of oral life [23,24]. Scholars have developed two short versions of 8 and 16 questions for the CPQ_11–14_ [25].

### 2.5. Data Analysis

The present study used IBM-SPSS Statistics version 25 (IBM, Armonk, NY, USA) to perform the statistical analysis of the study results. We report the obtained results as mean values and standard deviations. This study used the Student’s t-test and analysis of variance (ANOVA) to analyze the results. This study considered the results to be significant when *p* < 0.05 and highly significant when *p* < 0.01.

## 3. Results

### 3.1. Characteristics of Participants

Regarding the sex of the participants, girls (*n* = 73, 52.1%) outnumbered boys (*n* = 6, 47.9%). No statistically significant difference existed between the two groups. In relation to age, no statistically significant differences existed between the age groups: 11 (*n* = 40, 28.6%), 12 (*n* = 51, 36.4%), 13 (*n* = 29, 20.7%), and 14 (*n* = 20, 14.3%). The age distribution was very similar according to sex: 28.4 vs. 28.8%; 34.3 vs. 38.4; 20.9 vs. 20.5%; and 16.4 vs. 12.3% for girls and boys, respectively; *p* = 0.904.

### 3.2. Orthodontic Treatment Need Assessment

This study observed that 34.3% (*n* = 48) of the participants had a normal or mild malocclusion (Grade 1–2), which did not need orthodontic treatment, and 27.9% (*n* = 39) had a moderate malocclusion that required limited treatment (Grade 3). The remaining 37.8% (*n* = 53) needed treatment because of a severe or very severe malocclusion (Grade 4–5). Thus, most participants (62.2%) had a normal or mild malocclusion requiring no treatment or limited treatment.

### 3.3. Oral Health-Related Quality of Life Assessment

The present study found statistically significant differences in all categories of the CPQ-Esp_11–14_ questionnaire and in the questionnaire’s total score. The OHRQoL category with the highest score was social welfare (15.72 ± 1.91), followed by emotional well-being (11.92 ± 1.44). The category with the lowest score was oral symptoms (7.64 ± 1.39). The above indicates that the need for orthodontic treatment had the most negative impact on social and emotional well-being (Table 1).

This study concluded that sex did not have a statistically significant influence on OHRQoL (Table 2).

Age had a statistically significant influence on OHRQoL. Concerning oral symptoms (*p* < 0.01), functional limitations (*p* < 0.05), and the total score of the CPQ-Esp_11–14_ questionnaire (*p* < 0.05), statistically significant differences existed according to age. In both categories and the total score, the 11-year-old patient group reported the most negative impact of the need for orthodontic treatment on their OHRQoL (oral symptoms: 8.15 ± 1.42; functional limitation: 11.88 ± 1.62; total score: 48.25 ± 5.78). In all age groups, oral symptoms received the lowest score, but social well-being was the most negatively affected category.

Regarding oral symptoms, the 11-year-olds scored higher (8.15 ± 1.42) compared to the 14-year olds (6.95 ± 1.10). Concerning functional limitations, the 11-year-olds reported the most negative impact of the need for orthodontic treatment on their oral quality of life (11.88 ± 1.62) compared with the 13-year-olds, who experienced the fewest effects (10.90 ± 1.42).

Concerning the total score of the CPQ-Esp_11–14_ questionnaire, the 11-year-olds had the highest total score (48.25 ± 5.78) compared with the 13-year-olds, who had the lowest total score (44.97 ± 4.26) (Table 3). Based on these results, we concluded that the need for orthodontic treatment impacted OHRQoL most negatively in younger patients.

In another analysis, the present study evaluated age by grouping the total sample into two groups: 11–12 years and 13–14 years. Regarding oral symptoms (*p* < 0.01), functional limitations (*p* < 0.05), and the total score of the questionnaire (*p* < 0.05), the younger group scored significantly higher on emotional well-being (*p* < 0.05) as expected, whereas the group of patients aged 13–14 scored lower. As Table 4 shows, the younger patients scored higher than the older patients, which means they had a worse overall OHRQoL compared with the older patients.

### 3.4. Correlation between Orthodontic Treatment Need and Oral Health-Related Quality of Life

When analyzing how the severity of the need for orthodontic treatment influences OHRQoL, we observed statistically significant differences (*p* < 0.01) in all categories of OHRQoL and in the total score of the CPQ-Esp_11–14_ questionnaire.

The patients with the highest score on the questionnaire had the most severe malocclusion (Grade 5 in the IOTN) and the lowest OHRQoL.

Patients with a lower malocclusion severity (grades 1–2 and 3 in the IOTN) scored lower for oral symptoms and functional limitations. The category with the highest score (i.e., the worst impact on OHRQoL) was social well-being in patients with a very severe malocclusion (grade 5) (17.80 ± 1.63).

The average score for all the variables increased with the IOTN grade, especially from grade 4. Thus, the greater the need for orthodontic treatment (the higher the grade in the IOTN), the worse the OHRQoL (Table 5).

Based on the results of this study, we conclude that the need for orthodontic treatment has a highly significant negative impact on OHRQoL.

## 4. Discussion

The authors designed this study to be representative of the asthmatic child population of Salamanca, Spain, and one of its main strengths is the large sample of child patients within the required age range. This is the first study (to our knowledge) where the authors examined how treatment need influences OHRQoL in a child asthmatic population aged 11–14 years.

Scholars have developed several indices to identify the need for orthodontic treatment in a given population. The most prominent are the Dental Aesthetic Index (Dai); the Decayed, Missing, Filled Teeth Index (DMFI); and the Index of Orthodontic Treatment Need (IOTN) [26,27,28]. Different instruments to analyze quality of oral life in the pediatric population exist, each of which have been validated and used on participants of different age ranges; additionally, each are composed of a different number of items. The most prominent are the Child Perception Questionnaire (CPQ), the Child Oral Impact Daily Performance (Child-OIDP), the Child Oral Health Impact Profile (COHIP), and the Early Childhood Oral Health Impact Scale (ECOHIS) [23,24,29,30,31]. In the present study, we used the CPQ because it is one of the most used questionnaires in orthodontic studies on children and allows for the comparison of these results with those described by similar published studies. Two versions of the CPQ exist, one for 8–10 year-olds (CPQ_8–10_) and one for 11–14 year-olds (CPQ_11–14_) [23,24]. To analyze the quality of oral life and the need for orthodontic treatment, we used the Spanish version of the Child Perception Questionnaire (CPQ-Esp_11–14_) and the Index of Orthodontic Treatment Need (IOTN). Both questionnaires have been used in different studies [19,20,32,33,34,35,36].

In most cases, scholars developed the instruments used to measure OHRQoL as a self-administered questionnaire, and they are composed of a series of questions covering different aspects of OHRQoL [37]. Researchers recommend using short versions of the CPQ_11–14_ questionnaire, which have been validated, and scholars have concluded that they are also valid tools for quantifying OHRQoL in children [38,39,40,41].

The dental health component is a valuable tool for determining the priority of treatment needs in orthodontic care. This component is justified in that the more a patient deviates from an occlusion that is considered physiological, the greater the oral health problems. We used the IOTN to assess the malocclusion degree and to evaluate its aesthetic impact. In addition, when developing the IOTN, scholars solely based it on the opinion of orthodontic specialists; therefore, the IOTN is only adjusted to the objective need for orthodontic treatment based on the analysis of professionals [42,43,44].

In this study, we observed that 62.2% of the children did not need orthodontic treatment, whereas the remaining percentage, 37.8%, did. We found that age and sex were not influential factors, which Bellot-Arcís et al. [45] also found in Valencia, Spain. The authors reported that a high percentage of the sample, 80.8%, did not need orthodontic treatment according to the IOTN. However, the authors developed this study in an adult population without asthmatic pathology. Iranzo-Cortés et al. [46] also conducted a study to assess the need for orthodontic treatment in Valencia, Spain. They evaluated children aged 12 to 15, and as with the 12-year-olds in our study, they observed no significant influence of sex. In this case, 84.8% did not need orthodontic treatment.

When analyzing OHRQoL using the CPQ_11–14_ questionnaire, statistically significant differences regarding age existed. The results of a study conducted by Ydoğan [47] showed that, regarding emotional well-being, girls described a worse impact of the need for orthodontic treatment on their OHRQoL (girls: 13.3 ± 8.6; boys: 11.0 ± 7.4). Regarding the other categories, the author found no significant differences. Concerning the total score of the CPQ_11–14_ questionnaire, this author reported lower scores (girls: 40.1 ± 20.0; boys: 38.7 ± 19.9) compared with the present study (girls: 46.96 ± 5.15; boys: 46.40 ± 4.93). Additionally, this author found that emotional well-being had the highest score, whereas we found that social well-being had the highest score.

In a 2013 study, Scapini et al. [48] also evaluated the impact of malocclusions on OHRQoL in patients aged between 11 and 14 years old. Using the CPQ_11–14_ questionnaire, they found that the category with the highest score was oral symptoms (3.7 ± 2.4), and social welfare scored the lowest (2.5 ± 2.6). In our study, the results were contrary: social welfare (15.7 ±1.91) had much more impact compared with oral symptoms (7.64 ± 1.39), which had the least impact. When analyzing the correlation between malocclusions and OHRQoL, these authors reported a significant influence only on emotional and social well-being; in our study, we observed an influence on all of the questionnaire’s categories. This previous point can be partly justified by the fact that we used different indices to analyze the need for orthodontic treatment. These authors did find differences when analyzing sex: the girls indicated a worse impact of the need for orthodontic treatment on their OHRQoL. This finding is opposite to what we found, but we observed a trend in which girls described a more negative impact for most categories on the CPQ_11–14_ questionnaire (except for emotional well-being) and on the total score of the questionnaire, but these differences were not statistically significant. Moreover, these authors did not describe any statistically significant influence of age on the results.

Bittencourt et al. [49] described that, regarding oral symptoms, functional limitations, emotional well-being, and the total score of the CPQ_11–14_ questionnaire, girls described a worse impact of the need for orthodontic treatment on their OHRQoL. When analyzing the influence of age, these authors did not report significant differences, which we also found.

Ydoğan [47] reported a statistically significant correlation between the negative effect that the need for orthodontic treatment had on emotional and social well-being and the total score of the CPQ_11–14_ questionnaire. This corresponds with our findings and the findings of other authors [50,51].

Kragt et al. [52] also concluded that evidence of a clear inverse association between the severity level of a malocclusion and OHRQoL in child patients exists. They concluded, based on the results of previously published studies, that children aged between 11 and 14 are most likely to have some negative effects from malocclusions on their OHRQoL. Additionally, they concluded that no association between malocclusions and OHRQoL existed in younger age groups. Compared with what was previously described, Kolawole et al. [53] reported no significant influence of malocclusions on the OHRQoL of children, who also used the CPQ_11–14_ questionnaire. Based on the results described in the present study and according to the current scientific literature [32,52], an inverse proportional relationship exists between the severity of a malocclusion and the impact it has on the OHRQoL of patients.

The heterogeneity between the results of the published studies is explained, in part, by the method that each author used when evaluating the malocclusions, the age of the population, and the region or country in which the authors conducted the study.

The limitations of this study are that we performed it on a specific age group, which could affect the generalization of the results. The present study evaluated children aged 11 to 14 because scholars have only validated the CPQ_11–14_ questionnaire for this age group. Evaluating infant patients in a greater age range would be informative. Another limitation of this study is that we analyzed patients from only a single center; a replication of this study in clinical samples of adolescents in different centers could be beneficial.

These results contribute to understanding the importance of treating dental malocclusions in youth. The results of the objective need for orthodontic treatment may provide a reference for orthodontic care planning in adolescent patients with asthma. According to the scientific literature, no similar study has been conducted. The data from this study cannot confirm that childhood asthma is a risk factor for the presence or worsening of malocclusions. To analyze the influence of asthma on the variables explored in this study, researchers should conduct a case-control study and analyze whether this pathology has a considerable influence on the need for orthodontic treatment and OHRQoL. The use of questionnaires to evaluate OHRQoL in patients can increase the OHRQoL of patients who are going to start their treatment, based on what has been described in previous studies.

In the future, researchers should focus on populations of different ages and assess the influence of personal factors other than sex, age, and socioeconomic status when associating the need for orthodontic treatment with OHRQoL.

## 5. Conclusions

In the child asthma population studied, age and sex did not influence the need for orthodontic treatment. This study found that age had a significant influence on OHRQoL, as younger patients described a more negative impact of the need for orthodontic treatment on their OHRQoL via the CPQ_11–14_ questionnaire. Social well-being was the category in the CPQ_11–14_ questionnaire that had the greatest impact, whereas oral symptoms received the lowest score. The present study concluded that adolescent asthmatic patients aged 11–14 years old with more severe malocclusions will experience a worse impact on their OHRQoL.

## Figures and Tables

**Table 1 children-10-00176-t001:** Exploratory and descriptive analysis. Variables of the CPQ-Esp_11–14_ questionnaire on OHRQoL (*n* = 140).

CPQ-Esp_11–14_ Variable	Centrality		Range (Minimum/Maximum)	Variability	
	Mean	Medium		Standard Deviation	Interquartile Range
Oral symptoms	7.64	8.00	6/11	1.39	2.00
Functional limitation	11.41	12.00	8/15	1.40	2.00
Emotional well-being	11.92	12.00	8/16	1.44	2.00
Social well-being	15.72	15.00	11/20	1.91	1.75
Total score	46.69	45.00	36/60	5.03	7.00

**Table 2 children-10-00176-t002:** Inferential analysis. Variables of the CPQ-Esp_11–14_ questionnaire on OHRQoL according to sex (*n* = 140).

CPQ-Esp_11–14_ Variable	Mean (Standard Deviation)		Student’s *t*-Test		Effect Size: R^2^
	Boys (*n* = 67)	Girls (*n* = 73)	*t*-Value	*p*-Value	
Oral symptoms	7.48 (1.41)	7.78 (1.37)	1.29 ^NS^	0.198	0.012
Functional limitation	11.27 (1.30)	11.55 (1.48)	1.18 ^NS^	0.239	0.010
Emotional well-being	11.97 (1.40)	11.88 (1.48)	0.38 ^NS^	0.703	0.001
Social well-being	15.69 (1.74)	15.75 (1.88)	0.22 ^NS^	0.828	0.000
Total score	46.40 (4.93)	46.96 (5.15)	0.65 ^NS^	0.516	0.003

^NS^ = not significant (*p* > 0.05).

**Table 3 children-10-00176-t003:** Inferential analysis. Variables of the CPQ-Esp_11–14_ questionnaire on OHRQoL according to age (*n* = 140).

Categories	Mean (Standard Deviation)				ANOVA Test		Effect Size: R^2^
	11 Years (*n* = 40)	12 Years (*n* = 51)	13 Years (*n* = 29)	14 Years (*n* = 20)	*F*-Value	*p*-Value	
Oral symptoms	8.15 (1.42)	7.71 (1.36)	7.28 (1.33)	6.95 (1.10)	4.45 **	0.005	0.089
Functional limitation	11.88 (1.62)	11.37 (1.18)	10.90 (1.42)	11.35 (1.18)	2.92 *	0.037	0.060
Emotional well-being	12.23 (1.54)	12.00 (1.52)	11.52 (1.06)	11.70 (1.42)	1.58 ^NS^	0.196	0.034
Social well-being	15.80 (2.02)	15.80 (1.88)	15.28 (1.62)	15.60 (1.39)	0.96 ^NS^	0.411	0.021
Total score	48.25 (5.78)	46.88 (4.84)	44.97 (4.26)	45.60 (4.16)	2.86 *	0.039	0.059

^NS^ = not significant (*p* > 0.05); * = significant (*p* < 0.05); ** = highly significant (*p* < 0.01).

**Table 4 children-10-00176-t004:** Inferential analysis. Variables of the CPQ-Esp_11–14_ questionnaire on OHRQoL according to age (*n* = 140).

Categories	Mean (Standard Deviation)		Student’s *t*-Test		Effect Size: R^2^
	11–12 Years Old	13–14 Years Old	*t*-Value	*p*-Value	
Oral symptoms	7.90 (1.40)	7.14 (1.24)	3.18 **	0.002	0.068
Functional limitation	11.59 (1.41)	11.08 (1.34)	2.09 *	0.038	0.031
Emotional well-being	12.10 (1.53)	11.59 (1.21)	2.01 *	0.046	0.028
Social well-being	15.89 (1.94)	15.41 (1.53)	1.51 ^NS^	0.134	0.016
Total score	47.48 (5.29)	45.22 (4.18)	2.58 *	0.011	0.046

^NS^ = not significant (*p* > 0.05); * = significant (*p* < 0.05); ** = highly significant (*p* < 0.01).

**Table 5 children-10-00176-t005:** Inferential analysis. Variables of the CPQ-Esp_11–14_ questionnaire based on the IOTN (*n* = 140).

Categories	Mean (Standard Deviation)				ANOVA Test		Effect Size: R^2^
	Grade 1–2 (*n* = 48)	Grade 3 (*n* = 39)	Grade 4 (*n* = 28)	Grade 5 (*n* = 25)	*F*-Value	*p*-Value	
Oral symptoms	7.08 (1.01)	7.00 (0.95)	8.04 (1.23)	9.24 (1.39)	26.59	0.000 **	0.370
Functional limitation	10.96 (1.18)	10.85 (1.31)	11.75 (1.14)	12.80 (1.16)	16.86	0.000 **	0.271
Emotional well-being	11.08 (0.77)	11.31 (1.22)	12.71 (0.98)	13.60 (1.23)	40.78	0.000 **	0.474
Social well-being	14.75 (1.30)	14.97 (1.18)	16.57 (1.55)	17.80 (1.63)	33.94	0.000 **	0.428
Total score	43.87 (2.68)	44.13 (3.22)	49.07 (3.90)	53.44 (4.30)	55.30	0.000 **	0.550

** Statistically significant at *p* < 0.01.

## Data Availability

The data presented in this study are available upon request from the corresponding author.

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
