# Peer review of "Assessment of Orthodontic Treatment Need and Oral Health-Related Quality of Life in Asthmatic Children Aged 11 to 14 Years Old: A Cross-Sectional Study"

_children, 2023, doi:10.3390/children10020176_

Round 1

Reviewer 1 Report

the introduction is brief but does not clearly explain the importance of this study

In material and methods, the calculation of the sample should be followed by ethical aspects.

It is mentioned that a validated quality of life questionnaire is used and immediately it is said that two questionnaires were used?

table 1 with the Kolmogorov–Smirnov analysis is not necessary

The tables are not clear, data should be placed as absolute/relative frequencies. And although in material and methods it is indicated how the chosen instruments are composed, the quantitative method of measurement is not clarified

Reviewer 2 Report

Assessment of Orthodontic Treatment Need and Oral-Health-Related Quality of Life in Asthmatic Children Aged 11 to 14 Years Old: A Cross-Sectional Study is a well-written and constructed paper. I recommend minor corrections before accepting, and they are. 

1) Study inclusion criteria need to be elaborated, especially the asthma part. 

2) Who were the assessors? operator calibration and intra-class correlation coefficient (ICC) must be considered before reporting the results.

3) in the conclusion section, the authors report an inverse relationship, what do they mean by this? elaborate. 

Round 2

Reviewer 1 Report

The authors answered all suggestions